# New Aspects Towards a Molecular Understanding of the Allicin Immunostimulatory Mechanism via Colec12, MARCO, and SCARB1 Receptors

**DOI:** 10.3390/ijms20153627

**Published:** 2019-07-24

**Authors:** Vlad Al. Toma, Adrian Bogdan Tigu, Anca D. Farcaș, Bogdan Sevastre, Marian Taulescu, Ana Maria Raluca Gherman, Ioana Roman, Eva Fischer-Fodor, Marcel Pârvu

**Affiliations:** 1Faculty of Biology and Geology, Department of Molecular Biology and Biotechnologies, Babeș-Bolyai University, 400028 Cluj-Napoca, Romania; 2Institute of Biological Research Cluj-Napoca, branch of NIRDBS Bucureşti, 400113 Cluj-Napoca, Romania; 3Department of Molecular and Biomolecular Physics, National Institute for R&D of Isotopic and Molecular Technologies, 67-103 Donat, 400293 Cluj-Napoca, Romania; 4“Medfuture”-Research Centre for Advanced Medicine, University of Medicine and Pharmacy “Iuliu Haţieganu” 400349 Cluj-Napoca, Romania; 5Faculty of Veterinary Medicine, University of Agricultural Sciences and Veterinary Medicine, 400372 Cluj-Napoca, Romania; 6Faculty of Physics, Babeș-Bolyai University, 1 Kogălniceanu, 400084 Cluj-Napoca, Romania; 7Institute of Oncology “Ion Chiricuță”, 400015 Cluj-Napoca, Romania

**Keywords:** allicin, immunoglobulins, scavenger receptors, mechanism

## Abstract

The allicin pleiotropic effects, which include anti-inflammatory, anti-oxidant, anti-tumoral, and antibacterial actions, were well demonstrated and correlated with various molecular pathways. The immunostimulatory mechanism of allicin has not been elucidated; however, there is a possible cytokine stimulation from immunoglobulin release caused by allicin. In this study, when Wistar female rats and CD19+ lymphocytes were treated with three different doses of allicin, immunoglobulins, glutathione, and oxidative stress markers were assayed. Molecular docking was performed between S-allylmercaptoglutathione (GSSA)—a circulating form of allicin in in vivo systems formed by the allicin interaction with glutathione (GSH)—and scavenger receptors class A and B from macrophages, as well as CD19+ B lymphocytes. Our data demonstrated a humoral immunostimulatory effect of allicin in rats and direct stimulation of B lymphocytes by S-allyl-mercapto-glutathione, both correlated with decreased catalase (CAT) activity. The molecular docking revealed that S-allyl-mercapto-glutathione interacting with Colec12, MARCO (class A), and SCARB1 (class B) scavenger receptors in in vitro tests demonstrates a direct stimulation of immunoglobulin secretion by GSSA in CD19+ B lymphocytes. These data collectively indicate that GSSA stimulates immunoglobulin secretion by binding on scavenger receptors class B type 1 (SCARB1) from CD19+ B lymphocytes.

## 1. Introduction

The genus *Allium* exhibits a series of phytochemical properties attributed to the presence of organosulfur compounds such as alliin, allicin, ajoene, dithiins, and allyl sulfides (e.g., S-propenylcysteinesulfoxide). The existence and potency of *Allium*’s bioactive constituents vary with respect to its mode of preparation and extraction [1]. Organosulfur compounds present in *Allium* plants, either lipid or water-soluble form, are considered responsible for the beneficial effects of these herbs [2]. In terms of pharmacotherapy, allicin is the common name of the allicin-related compounds which include pure allicin in various concentrations. In vivo experiments regarding the effects of allicin use Allimed^®^ (Allicin International Ltd. (East Sussex, UK) or Allimax^®^ (Allicin International Ltd., East Sussex, UK) [3,4,5], which has developed a patented process for providing biologically active and stabilized forms of allicin when mixed with its derivatives. In humans and animals, allicin acts by decreasing extracellular concentration of sulfhydryl groups [6], reacting with glutathione in order to recover the normal oxidative status, or by generating S-allyl-mercapto-glutathione [7,8]. When allicin and its derivatives were tested in different pathologies as adjuvants or therapeutic compounds, the authors mentioned antioxidant [9], anti-inflammatory [7,10], antitumoral [11], glycostat regulatory [7,12], antibiotic [13], and immunostimulatory effects [6]. However, the mechanism through which allicin stimulates the humoral immune response is not well understood. Several experiments revealed that allicin and *Allium* extracts enhance macrophage production of TNFα and NO in a dose-dependent manner, thereby stimulating cytokine secretion, phagocytosis promotion, and an increasing production of intestinal IgA [7,14]. Furthermore, in vitro tests pointed out that allicin inhibited ICAM-1 expression in endothelial cells [15] and JNK/p38 in Caco-2 cells [16]. Zhang and coworkers [17] noticed JNK signaling suppression by allicin in the context of cognitive impairment in APP/PS1 mice. The allicin effect was further described in induced-pathological scenarios, such as antioxidant and anti-inflammatory by scavenging free radicals [18], antitumoral through JNK stimulation or Nrf2 inhibition [19], glycostat modulator by hexokinase and enolase inhibition [20], and antibiotic through the inhibition of bacterial Cys-proteinase, thioredoxin-reductase, or alcohol-dehydrogenase [21]. However, the studies mentioning the immunostimulatory effects of the allicin did not describe the mechanism through which allicin stimulated the immunoglobulin production. Arreola et al. [22] mentioned that garlic enhances the functioning of the immune system through mechanisms including the modulation of cytokine secretion, immunoglobulin production, phagocytosis, and macrophage activation. Moreover, allicin induced the activation of ERK1/2 which belongs to the mitogen-activated protein kinase family leading to the activation of macrophages. Furthermore, Kang and coworkers [14] claimed that the immunostimulatory activity of allicin may be mediated through the upregulation of secretory molecules in macrophages and that it also plays a role in triggering the activation of these cells. However, the authors did not mention the receptors and the signaling pathways for allicin-stimulated immunoglobulin secretion. The involvement of B cell receptors in allicin-stimulated immunoglobulin secretion is not a possible pathway according to Leadbetter et al. [23] and Janeway et al. [24]. They [23,24] proposed a few ligands for B cell receptors, such as viral capsid, extracellular matrix, bacterial capsule, chromatin-IgG complex, or proteins from necroptosis, apoptosis, or hypoxic tissues. Based on the experimental data, the macrophages seem to play a key role in allicin-stimulated immune response as well as in immunoglobulin production after allicin exposure, but a definite mechanism has not been suggested. Our assumption is based on the evidence presented by Borlinghaus et al. [7] and Rabinkov et al. [8], who described the interaction of allicin with GSH in order to form S-allylmercaptoglutathione (GSSA). We claim that the GSSA molecule has an agonistic effect like polyanionic molecules (i.e., modified low density lipoproteins or polysaccharides) on scavenger receptors class A (Colec12 and MARCO) and B1 exhibited by macrophages and B lymphocytes, and that this generates B cell stimulation, via a nonspecific humoral immune response. As was noticed by Ferreira and coworkers [25], the scavenger receptors are frequently involved in the nonspecific humoral immune stimulation. Therefore, the purpose of the current study was to establish the function of the scavenger receptors in immunostimulation by the allicin.

## 2. Results

### 2.1. Blood Oxidative Stress Is Related to Immunoglobulin Secretion in a Dose-Dependent Manner after Allicin Treatment

Organosulfur compounds play a dual role considering their antioxidant as well as pro-oxidant properties. Following allicin administration, our data exhibit a prominent antioxidant effect of allicin, marked by a decrease in catalase (CAT) activity (Figure 1A) in comparison with the control group. Significant lower CAT activity (ANOVA test, *p* < 0.001) was confirmed by Bonferroni’s post hoc test in A1 (1.25 mg/kg), 38.2 ± 2.33 U/mL, A2 (2.5 mg/kg), 31.9 ± 3.50 U/mL, and also in A3 (5 mg/kg), 26.7 ± 4.53 U/mL. One-way ANOVA also revealed an interesting significant relation (*p* < 0.05) between A3 and superoxide radicals by an increase in superoxide dismutase (SOD) activity (Figure 1B) (C, 0.77 ± 0.004 USOD/mL/min; A3, 0.93 ± 0.02 USOD/mL/min). In the C group, IgA concentration (Figure 2A) was 20.8 ± 2.17 mg/dL, whereas the administration of allicin increased IgA concentration; thus, A1 was 44.8 ± 0.75 mg/dL; A2, 46 ± 1.22 mg/dL; and A3, 49.8 ± 1.40 mg/dL. Moreover, IgG (Figure 2B) in C was 160 ± 10.60 mg/dL and allicin increased IgG; thus, in A1, IgG was 158.3 ± 6.4 mg/dL; in A2, 166 ± 11.34 mg/dL; and in A3, 188.8 ± 6.22 mg/dL (*p* < 0.001). IgM (Figure 2C) followed the same ascendant trend as IgA and IgG, and the ANOVA test revealed significant variations between the control and experimental groups (ANOVA test, *p* < 0.001) for IgM as well as IgG. IgM concentration in C was 23.3 ± 2.01 mg/dL; in A1, 47.8 ± 1.31 mg/dL; in A2, 52.0 ± 1.47; and in A3, 50.8 ± 2.11 mg/dL (*p* < 0.001). Albumin (ALB) concentrations (Figure 2D) (normal range: 37–58 g/L) were slightly elevated after allicin treatment compared to the control group (ANOVA test, *p* < 0.05) but without clinical significance. In C, ALB was 29.2 ± 1.31 g/L; in A1, 33.2 ± 1.13 g/L; in A2, 34.7 ± 0.94 g/L; and in A3, 37.8 ± 1.6 g/L (*p* < 0.05). The albumin concentration in the control rats was lower than the reference range because in rats, albumin varies with age, hormonal fluctuations. as well as the degree of hydration [26,27]. Allicin increased the albumin concentration with values that remained in the normal range. Total proteins (TP) (normal range: 5.6–7.6 g/dL) (Figure 2E) (C, 7.4 ± 0.47 g/dL; A1, 11.6 ± 0.82 g/dL; A2, 13.3 ± 1.13 g/dL; A3, 13.5 ± 1.17 g/dL) are elevated after allicin administration which is confirmed by both ANOVA (*p* < 0.01) and Bonferroni’s post-hoc test (*p* < 0.001).

The correlation test revealed that the downregulation of oxidative stress is related to enhanced immunoglobulin secretion (IgA vs. CAT, R = −0.76, *p* < 0.05; IgM vs. CAT, R = −0.71, *p* < 0.05; IgG vs. CAT, R = −0.85, *p* < 0.01). In vivo experimental data demonstrated that allicin enhanced the immunoglobulin’s secretion without B cell count changes (Table 1) with a *p*-value > 0.05 in all white blood cell (WBC) variations.

### 2.2. Serum GSH Concentration Was Decreased after Allicin Administration by Forming S-Allylmercaptoglutathione (GSSA)

The most abundant non-protein thiols that can potentially interact with allicin are reduced glutathione (GSH) and cysteine. S-allylmercaptocysteine is known already as the product formed when cysteine reacts with allicin [28]. However, our data cannot be justified by allicin’s interaction with cysteine because the Ellman reagent (5,5’-dithio-bis-[2-nitrobenzoic acid]), which was used in the GSH assay, reacts with an extended molecular cluster which contains all sulfhydryl groups. Rabinkov and coworkers [8] demonstrate that in an in vivo system, allicin reacts spontaneously 1:1 with reduced glutathione and generates S-allylmercaptoglutathione (GSSA), a compound which does not react with the Ellman reagent. The concentration of GSH in blood is about 100-fold higher than cysteine; therefore, GSH remains the main candidate for the in vivo interaction with allicin. One-way ANOVA (*p* < 0.001) and Bonferroni’s post-hoc test (*p* < 0.001) show a significant decrease in GSH concentration (Figure 2F) when it comes to animals treated with allicin (C, 113.3 ± 6.87 nmol/g protein; A1, 75.5 ± 7.17 nmol/g protein; A2, 60.7 ± 7.52 nmol/g protein; A3, 58.3 ± 7.51 nmol/g protein).

### 2.3. CD19+ Lymphocytes Were Stimulated by GSSA (GSH–Allicin) Exposure

The in vivo analyses demonstrate that allicin increases immunoglobulins and decreases catalase activity in serum. However, these data do not elucidate the mechanism of immunostimulation, considering that there could be possible involvement of both macrophages and B cells in the increase in immunoglobulins. To exclude the macrophages’ contribution in the observed immunostimulation effect of the allicin, we tested the B lymphocyte reaction after allicin and allicin + GSH exposures. CD19+ lymphocytes exposed to allicin (A) and GSH + allicin (GA) showed the same reaction pattern as in vivo models in terms of oxidative stress and immunoglobulin secretion. Allicin and GSH + allicin decreased CAT activity in CD19+ cell cultures (ANOVA test, *p* < 0.001) (Figure 3A). In C, CAT was 0.81 ± 0.11 U/mL; in A1, 0.30 ± 0.03 U/mL; in A2, 0.29 ± 0.04 U/mL; in A3, 0.18 ± 0.03 U/mL; in GA1, 0.13 ± 0.02 U/mL; in GA2, 0.09 ± 0.01 U/mL; and in GA3, 0.07 ± 0.01 U/mL (*p* < 0.001). Moreover, GSH reduced CAT activity at 0.49 ± 0.07 U/mL (*p* < 0.001). SOD activity (Figure 3B) was decreased (ANOVA test, *p* < 0.01) after allicin and allicin + GSH treatment (*p* < 0.05), and the most prominent decrease was observed in GA2 and GA3 (*p* < 0.001). In C, SOD activity was 1.21 ± 0.11 USOD/mL/min; in A1, 0.91 ± 0.06 USOD/mL/min; in A2, 0.93 ± 0.07 USOD/mL/min; and in A3, 0.98 ± 0.05 USOD/mL/min. In GA1, SOD was 0.89 ± 0.05 USOD/mL/min; in GA2, 0.61 ± 0.10 USOD/mL/min; and in GA3, 0.50 ± 0.07 USOD/mL/min. GSH also decreased SOD activity at 0.99 ± 0.04 USOD/mL/min (*p* < 0.05).

As in in vivo systems, oxidative stress was correlated (IgM/CAT, R = −0.60, *p* < 0.05; IgG vs. CAT, R = −0.61, *p* < 0.05) to immunoglobulin secretion in GSH + allicin-treated cells with the exception of IgA (Figure 4A), which has not presented a significant variation (ANOVA test, *p* > 0.05, *p* > 0.05). IgG (in C, 33.3 ± 5.36 mg/dL) (Figure 4B) was not affected by GSH exposure (G group, 36.6 ± 2.40 mg/dL, *p* > 0.05) or by allicin exposure (A1, 34.3 ± 3.50 mg/dL; A2, 41.6 ± 5.78 mg/dL; A3, 42.7 ± 3.52 mg/dL, *p* > 0.05), but the combination of GSH + allicin (GSSA formation) determined a prominent dose-dependent increase of IgG (*p* < 0.001) in the GA1 (123.7 ± 8.01 mg/dL), GA2 (215.3 ± 7.63 mg/dL), and GA3 (262.3 ± 17.27 mg/dL) groups (ANOVA test, *p* < 0.001). GSH or allicin exposure did not induce the increase of IgM secretion (*p* > 0.05) (Figure 4C). The value of IgM in the control group was 12 ± 1.52 mg/dL, whereas in G group it was 12.7 ± 1.45 mg/dL. After allicin exposure, the IgM value in A1 was 16 ± 2.30 mg/dL; in A2, 16.3 ± 2.60 mg/dL; and in A3, 17 ± 3.21 mg/dL (*p* > 0.05). GSH + allicin treatment of the CD19+ lymphocytes induced an enhanced secretion of IgM (*p* < 0.001) in the GA1 (30 ± 3.05 mg/dL), GA2 (37 ± 1.52 mg/dL), and GA3 (62.7 ± 2.60 mg/dL) groups (ANOVA test, *p* < 0.001). Furthermore, TP (Figure 4D) were proportionally increased with immunoglobulins. In C, TP were 1.53 ± 0.08 g/dL, and in G, they were 1.56 ± 0.14 g/dL. Allicin treatment determined in A1, 1.66 ± 0.18 g/dL of the TP; in A2, 1.60 ± 0.11 g/dL; and in A3, 1.86 ± 0.08 g/dL (*p* > 0.05). GSH + allicin treatment induced a prominent protein secretion by the CD19+ lymphocytes. Thus, in GA1, the total proteins were 2.1 ± 0.09 g/dL; in GA2, 2.3 ± 0.17 g/dL; and in GA3, 3.1 ± 0.24 g/dL (*p* < 0.001) (ANOVA test, *p* < 0.001).

The correlation diagram (Figure 5A,B) between all the parameters investigated using the first two principal components of the principal component analysis (PCA) model was obtained after applying PCA for (A) in vivo and (B) in vitro parameters. As seen in Figure 5A, the serum catalase is negatively correlated and, on the right side of the correlation circle, immunoglobulin, TP, ALB, and SOD were clustered and strongly positively correlated.

In turn, in CD19+ cells, a strongly negative correlation was observed between CAT with respect to immunoglobulins, TP, ALB, or SOD. As compared with the correlation diagram obtained in vivo, there are two clusters for the in vitro parameters (Figure 5B), namely the antioxidant enzymes are on the left part of the circle, whereas the immunoglobulins and TP are on the right part; this indicates a negative correlation between them. CAT and SOD activities are negatively correlated with immunoglobulins, which demonstrates a functional relationship between oxidative stress and immunoglobulin secretion.

### 2.4. Molecular Docking Suggested the Most Specificity of S-Allylmercaptoglutathione (GSSA) Binding on Colec12, MARCO, and SCARB1 Scavenger Receptors

In addition to the previously mentioned results, we performed molecular docking on six ligand–receptor systems based on the non-specific immunostimulatory mechanism mentioned by Tzianabos [29] and Plüddemann [30] for polyanionic molecules, polysaccharides, or modified LDL. These authors noticed that macrophage scavenger receptors such as MARCO, Colec12, or Scar B1 could be responsible for the non-specific immunostimulatory action of various molecules. The binding energies between GSSA (constructed molecule can be seen in Figure 6) and scavenger receptors (SRs) lie within a narrow range of −4.7 kcal/mol to −6.7 kcal/mol (Table 2).

However, after analyzing the systems from both a geometric and energetic perspective, we observed that Msr1, SCARA3, and SCARA5 have low geometrical affinity and small interaction energy, so they were discarded from further analysis.

Colec12 (quadrimer), MARCO (dimer), and SCARB1 (monomer) have high interaction energy and good geometric match. Basically, these proteins host the ligand in a pocket when they are monomer (SCARB1, Figure 7C III) or they naturally self-assemble so as to create this pocket (Figure 7A III, B III).

The highest predicted affinity between GSSA and these receptors resulted for Colec12 (SCAR A). However, MARCO (SCARA) and SCARB1 (SCARB) are also very probable as an interaction mechanism, having very specific binding sites. The charge distribution on the surface of the three receptors is different, Colec12 presenting most of its surface as electronegative (Figure 7A IV), whereas MARCO (Figure 7B IV) and SCARB1 (Figure 7C IV) have smaller areas of either electronegative or electropositive charge, most of their surfaces being neutral. This is the reason why GSSA binds more strongly to Colec12 (with a binding energy of −6.7 kcal/mol), compared to MARCO and SCARB1 (−5.7 kcal/mol). On the other hand, when the molecular docking of GSSA to MARCO was performed, all 90 conformers bound to the same site (Figure 7B II), and a pocket formed between the two monomers of the MARCO receptor. In the case of SCARB1, GSSA is somehow trapped in the “insides” of the receptor and this binding is based on a more geometrical match rather than a chemical or electrostatic one. However, SCARB1 is a candidate for the GSSA lymphocyte stimulation mechanism based on the experimental data obtained from the in vitro tests.

## 3. Discussion

The antioxidant properties of allicin were well demonstrated by using in vitro and in chemico methods. DPPH, Trolox, or hemoglobin-based antioxidant assays demonstrate in different ways the antioxidant effects produced by allicin, according to Moţ et al. [32]. The antioxidant properties of allicin related to immune response is the main area of investigation in our study, which is also related to other research [8]. It is well known that allicin reacts with reduced glutathione and forms S-allylmercaptoglutathione (GSSA) [8], a new antioxidant compound which is not detectable through the Ellman method, whereas Salehi [6] has reported decreasing glutathione pool and sulfhydryl levels after allicin administration. In ex vivo experiments done with fetal brain slices under iron-induced oxidative stress, GSSA significantly lowered the production levels of lipid peroxides [7]. GSSA and allicin have a similar reaction, as both SH-modifiers and antioxidants, which implies that the thioallyl moiety plays a key role in the biological activity [8]. Thus, a prominent decreasing concentration of thiols after allicin administration in a dose-dependent manner is based on these reactions which maintain the antioxidant properties of the allicin. The oxidative equilibrium is also indicated by oxidative stress enzymes such as catalase (CAT) and superoxide dismutase (SOD) activity, which are frequently reported in a direct/indirect proportionality relationship, depending on the sampling moment of the experiment as well as experimental model [33]. Three days of neuropsychological stress increased blood CAT activity at 64 U/mL and seven days of the same stressor were associated with 67 U/mL of CAT, a non-significant increase, whereas CAT activity in the control group was 40 U/mL [34]. Native hemoglobin, in 12 h after i.v. injection, increased CAT at 280 U/mL, whereas the control rats showed 118 U/mL of CAT activity [35]. In turn, Sagor et al. [36] noticed a decrease in CAT activity in plasma after CCl4 administration at 2.9 U/min, whereas in the control group, the enzyme was 6 U/min. These values depict that the increase or decrease in CAT activity should be discussed only in an experimental context of the values. Our data show catalase decreasing activity after allicin treatment in an inverse dose relationship manner, ranging between 38 U/mL (A1) and 26 U/mL (A3) as opposed to CAT in the control group, which was 40.3 U/mL. SOD was widely unchanged even if this enzyme gives substrate to CAT, which was decreased by allicin treatment. These data demonstrate that there is no obligatory proportional relationship between CAT and SOD and that increasing CAT does not necessarily mean an increased oxidative stress. Given this context, unchanging SOD and decreasing CAT suggest that superoxide radicals are not generated and peroxide decreases. In this context, decreasing CAT activity can be related to a lowered oxidative stress because CAT was decreased in both serum and cell cultures, whereas SOD was slightly increased in serum and decreased in cell cultures. Catalase is a common factor for our in vivo and in vitro tests and CAT decreasing was used as a single indicator of the antioxidant effect of the allicin. Furthermore, as a second-line antioxidant enzyme, CAT decreasing is subsequent to GPX intervention in peroxide scavenging [37], meaning there is an overall decrease in the oxidative stress. CAT and SOD decreased in their activities almost simultaneously; however, we noticed that these two enzymes (SOD and CAT) did not follow the same behavior as expected. SOD is an inducible and polymorphic enzyme that can be activated together with GPX by antioxidants via Nrf2 signaling [38]; therefore, SOD increasing in serum in the A3 group suggests that allicin in a high concentration could activate this enzyme. The apparent contradiction between serum SOD increasing and CAT decreasing reveals the polyfactorial feature of the antioxidant response in in vivo systems, whereas in CD19+ cells SOD and CAT were varied in tandem. SOD and CAT levels were decreased in CD19+ cells after 24 h of allicin and allicin-GSH exposures because superoxide production was balanced by the direct allicin scavenging effect [1,6,11]. It should also be noted that peroxide concentration was also reduced. Given this context, the major difference between serum and CD19+ cells is that in vivo systems utilize complementary antioxidant factors (nonenzymatic antioxidants, hormonal regulation), whereas CD19+ cell cultures are an isolated system. The oxidative balance was overall improved by allicin as well as the humoral mediated immunity; this was exemplified by the fact that immunoglobulins were increased, whereas WBCs were stationary and the oxidative stress decreased. The immunomodulatory properties of allicin and *Allium* extracts have been well documented for garlic [10,22]. However, the mechanism through which allicin and *Allium* extracts increased immunoglobulins is not well understood. Many studies noticed that garlic stimulates anti-inflammatory effects in vivo, whereas in vitro it stimulates pro-inflammatory cytokines such as IL-1 or TNF [39,40] and induces a humoral- and cellular-mediated immune response [22,41]. Studies with purified allicin also mentioned the same opposite effect on cancer cells, cultured WBCs, or fibroblasts [42,43], but all studies reported that this immunomodulatory effect lacked the explanation about the mechanism. In our in vivo study, immunoglobulins marked a prominent increase associated with stationary WBCs and decreasing oxidative stress. Moreover, CD19+ lymphocytes after GSH + allicin exposure (which spontaneously formed GSSA) [8] also showed an increased immunoglobulin secretion and decreased oxidative stress. These in vitro effects were not observed after a single allicin treatment, and this denotes a determinant role of GSH in allicin immunostimulatory effects by GSSA signaling. There are two possible pathways related to the increase of immunoglobulins: (i) cytokine pathway which has already been mentioned by some authors [22,39,44] and (ii) polysaccharide-like immunostimulation, according to Ferreira and coworkers [25], by scavenger receptors from macrophages and/or B cells, as nonspecific receptors related to immunoglobulin secretion [45]. Rahman [46] highlighted the result that garlic decreased the NF-κβ level, which is a central transcription factor and which has a central role in the expression of genes that control the immune response. The same author extrapolated the cytokine pathway as a basic mechanism of immunoglobulins’ increasing concentration. However, the stimulus that activates different cytokines which exert an immunostimulatory action in B lymphocytes was not mentioned. Our assumption is that allicin interacts with certain GSH and forms GSSA with a potent effect on the host immune response. Therefore, to a lesser extent, certain polyphenols could act in a similar manner with allicin. These hypothetic mechanisms are based on the verified fact that proteins, peptides, lipopolysaccharides, glycoproteins, or lipid derivatives have all been characterized by potent effects causing an immune response [25,29]. Furthermore, the same authors noticed that IgM is the main Ig fraction that is increased by antigens like glucans, manans, protein-bound polysaccharides, or hyaluronic acid. The present experimental data support their results; IgM and IgA were the most prominent increased Ig fractions. Our polysaccharide-like immunostimulation proposed mechanism of allicin immunostimulatory effect is further supported by the molecular docking. Arreola et al. [22] mentioned that fructooligosaccharides from garlic extract could stimulate Ig secretion by a similar polysaccharide-like mechanism. However, Allimed^®^ (Allicin International Ltd., East Sussex, UK) does not contain fructooligosaccharides and exerts a notable Ig stimulation. Certainly, cytokine pathway and polysaccharide-like immunostimulation are synergistic, but for allicin, the polysaccharide-like mechanism is the most likely pathway for humoral immunostimulatory effect. Molecular docking revealed that GSSA interacted by electrostatic and geometric match mechanisms with class A of the scavenger receptor type, Colec12 and MARCO, as well as with SCARB1 as an exponent of class B of the scavenger receptors which presented a high geometric specificity with GSSA and a moderate electrostatic interaction. Zhang et al. [17] and Jördo et al. [45] noticed that class A of the scavenger receptors is expressed only on macrophages from well-defined tissues, such as the marginal zone of the spleen, lymphatic nodules, or dendritic cells. Arredouani [47] mentioned a MARCO receptor recognizing ligands that are often polyanionic in nature as well as Colec12 which binds oxidized LDL and polyanionic molecules. Based on our experimental data and according to related studies [17,25], GSSA binds to Colec12 and MARCO receptors (class A) from macrophages which generate proinflammatory interleukins (IL-1, IL-8, IL-12) responsible for lymphocyte stimulation. In turn, T cells by IL-4, IL-5, and IL-6 stimulate B cells, thereby increasing immunoglobulin secretion. However, our data demonstrate that GSSA stimulates immunoglobulin secretion by direct effect on B lymphocytes, and generates a direct pathway of GSSA immunostimulation via class B1 of the scavenger receptors (as can be seen in Figure 8). According to other studies [45], B lymphocytes expressed the class B (SCARB1) of the scavenger receptors and major histocompatibility complex (MHC) - like molecule CD1d. SCARB1 was mentioned as also expressed on macrophages, but our data demonstrated that, in the absence of macrophages, B cells were stimulated. This suggests a macrophage-independent mechanism of Ig secretion after GSSA exposure. Molecular docking demonstrated that GSSA binds to SCARB1 receptors with −5.7 kcal/mol and very high geometric matching. In comparison, carbinoil stabilizes SCARB1 at −3.2 kcal/mol [48] and the LDL-derived pyrene-sphingomyelin binding energy was 7.1 ± 0.9 Kcal/mol [49]. SCARB1 receptors have been well described as a high-density lipoprotein receptor which mediates both the selective uptake of cholesterol esters and the efflux of cholesterol. Additionally, this receptor has recently been implicated in the recognition of other pathogens, estradiol, and vitamin E [50,51]. According to Neyestany et al. [52] and Ercal et al. [53], lycopene administration increased immunoglobulin levels, while the enhanced oxidative stress after lead exposure considerably decreased immunoglobulin concentration in blood. Data gathered from theoretical models associated with experimental findings indicate that GSSA-induced immunoglobulin secretion could occur through two pathways, namely, (i) indirect route via macrophage with class A SR type (Colec12, MARCO) and class B SR type 1 (SCARB1) and (ii) direct signaling, by interaction of GSSA with class B of SR type 1 (SCARB1) from B plasma lymphocytes which cross-talk with B cell receptor (BCR) signaling pathway for Ig secretion (Figure 8). Signaling through SCARB1 interferes with the B cell receptor (BCR) signaling pathway, a specific receptor involved in immunoglobulin secretion via tyrosine kinase LYN and SYK. However, SYK tyrosine kinase can direct the signaling cascade either to enable PI3K or activate ERK. The coactivation of PI3K–ERK is also a possible cross-talk signaling pathway of SCARB1–BCR [54,55] in B lymphocytes. In immunoglobulin secretion via BCR, LYN/SYK–ERK is the main signaling pathway, in comparison to the CD19+ cells where stimulation by allicin and GSSA was associated with PI3K activation [55]. SCARB1 signaling becomes overlapped with BCR signaling cascade in tyrosine kinase LYN-SYK, which determines the next reactions (via PI3K and ERK) for immunoglobulin secretion. The decrease of the oxidative stress maintained the immunoglobulin secretion and posttranslational folding.

## 4. Materials and Methods

### 4.1. In Vivo Studies

The experiment was designed in order to evaluate the dose-dependent effect of allicin (Allimed^®^, Rye, East Sussex, UK). Allicin was administered in three different doses. The animals were divided into four equal groups (five animals/group): control (C), 1.25 mg allicin/kg (A1), 2.5 mg allicin/kg (A2), and 5 mg allicin/kg (A3). The treatment was administered during 14 days, with daily oral administration via gavage.

### 4.2. Animals

Adult (two-month-old) Wistar female rats weighing 180 ± 20 g were provided ad libitum access to standard chow and water. Animals were maintained in a light- and temperature-controlled room with a light/dark cycle of 12 h/12 h under 23 °C constant temperature.

### 4.3. Ethics Statement

Animal care and procedures were carried out in accordance with Directive 2010/63/EU and national legislation (law no. 43, since 11 April 2014). The procedures of the current work were approved by the Ethical Committee of Babeș-Bolyai University (Institutional Review Board, decision no. 2012, project identification code: IZO-MOL-EA PN19 35 02 01, since 3 February 2016). The animals were euthanized by deep prolonged narcosis and were considered dead when no respiratory and heart activity was recorded.

### 4.4. Hematology

At the end of the experiment, 12 h after the last allicin administration, the animals were deeply anesthetized with isoflurane. For hematological evaluation, blood samples from retro-orbital plexus (200 µL) were collected in K-EDTA anticoagulant tubes, were labelled and immediately submitted for hematological analysis. Complete blood counts were performed using an Abacus Junior Vet automatic analyzer (Diatron Messtechnik, Vienna, Austria).

### 4.5. Immunoglobulins

Blood samples (2 mL/rat) for biochemical assay were drawn on clot-activated vacutainers and the serum was separated by centrifugation and analyzed for the determination of immunoglobulins (IgA, IgM, IgG) by immunoturbidimetric method with anti-IgA, anti-IgG, and anti-IgM specific antiserum [34,35]. Immunoglobulins were assayed in serum and CD19+ cell cultures.

### 4.6. Oxidative Stress, Total Proteins, and Albumin Assay

Blood biochemistry was done with serum that was initially separated by centrifugation. Serum was used for the determination of total proteins (TP), reduced glutathione (GSH), catalase activity (CAT) and superoxide dismutase activity (SOD) [34,35]. CD19+ cell cultures were used for the assay of TP, CAT, and SOD.

### 4.7. In Vitro Testing on CD19+ B Cells

Human peripheral blood mononuclear cells (PBMC) were isolated from fresh peripheral whole blood. PBMC isolation was performed through gradient separation. The blood was collected in Li-heparin-coated Vacutainer test tubes (from Beckton Dickinson and Co., Plymouth, UK), mixed 1:1 with phosphate-buffered saline solution, and disposed gently on Histopaque 1.077 layer (PBS and Histopaque from Sigma-Aldrich, St. Louis, MO, USA). After centrifugation at 2000 rpm, the lymphocyte layer (“ring”) formed between the plasma and the Histopaque layer was separated by aspiration; the cells were washed twice in ice-cold separation buffer containing 0.5% fetal calf serum (FCS, from Sigma-Aldrich) in phosphate-buffered saline solution. Aliquots from the whole population were subjected to magnetic separation. Lymphocytes were counted using a Bürker hemocytometer and then separated using the CD19 micro beads (kit provided by Miltenyi Biotec, Bergisch Gladbach, Germany), as described earlier [56]. Briefly, the PBMC cell suspension containing 107 cells/mL was incubated with the appropriate amount of antibody-labelled magnetic micro beads for variable time periods depending on the selected phenotype, and after washing in cold separation buffer, cells were positively separated on LS columns; the magnetic field was generated with Midi MACS magnets (from Miltenyi Biotech). The PBMC enriched in CD19-positive cells were resuspended in culture medium and seeded on 96-well microplates, at a concentration of 25,000 cells/well. The cell culture medium was RPMI-1640 (from Gibco, Grand Island, NY, USA) with 10% fetal bovine serum (FBS) (Gibco, Grand Island, NY, USA), 1% sodium pyruvate (Sigma-Aldrich, St. Louis, MO, USA), 1% glutamine (Sigma-Aldrich, St. Louis, MO, USA) and 1% penicillin-streptomycin 10,000UI/mL (Sigma-Aldrich, St. Louis, MO, USA). The cells were grouped as follows: control, GSH 1 mM (according to Losa, 2003), A1 (allicin 10 µg/mL), A2 (allicin 30 µg/mL), A3 (allicin 60 µg/mL), GA1 (GSH 1 mM + allicin 10 µg/mL), GA2 (GSH 1 mM + allicin 30 µg/mL), and GA3 (GSH 1 mM + allicin 60 µg/mL). In cell culture medium, allicin doses were related to in vivo allicin administration doses according to similar studies [57,58]. After 24 h, the supernatant was removed and immunoglobulins (IgA, IgG, IgM) and total proteins (TP) were assayed. Cells were then subjected to lysis using RIPA buffer at 4 °C during 12 h. After cell lysis, the samples were centrifuged at 14,000 rpm and the supernatant was used in order to assay CAT and SOD activity.

### 4.8. Molecular Docking

In this study, we used the AutodockVina program (version 4, Molecular Graphics Laboratory, The Scripps Research Institute, La Jolla, CA, USA) which employs a Lamarckian genetic algorithm and thus refines the myriad of possible docking geometries to the most probable one, with the highest binding energy between receptor and ligand. Because the algorithm is non-deterministic and every search may return different results, 9 binding modes were generated for each simulation and the operation was repeated 10 times for every system. This way, we obtained 90 docking conformations in each case, a number that we consider large enough to characterize the specificity of interactions between selected scavenger receptors and S-allylmercaptoglutathione (GSSA). To run the AutodockVina algorithm, we first constructed the systems in Autodock Tools 4 [59]. We allowed full flexibility for the ligand, while the receptor was kept rigid. Only the polar hydrogens were taken into account. The chemical structure of S-allylmercaptoglutathione was drawn in Gauss View Version 5 (Gaussian Inc. Wallingford, CT, USA,) [31], whereas the structures of the scavenger receptors (SRs) were used as retrieved from the Protein Data Bank. Specifically, we used several scavenger receptors of type A, such as Colec12 (PDB ID: 2OX9) [60], MARCO [61], Msr1 (PDB ID: 1BY2) [62], SCARA3 (PDB ID: 5CTD) [63], and SCARA5 [58], and one scavenger receptor of type B, SCARB1 [64]. The selection of these SR was done considering (i) the GSSA possible reaction with binding sites of the receptor and (ii) protein structure availability in databases. The GSSA interacts by its reactive groups (-S-CH2, -OH, -NH2, -COOH, -NH) with cysteine-rich domain (selected SR class A) as well as with N-linked glycosilation sites (SR class B). Moreover, class A and B of the SR were mentioned as the main protein cluster involved in non-specific humoral immune response stimulation [45].

### 4.9. Allicin and Other Chemicals

Stabilized allicin formulation as Allimed^®^ was purchased from Allicin International Ltd. (East Sussex, UK). The reagents included in standard assay packets with colorimetric and kinetic methods were obtained from BioMaxima (Lublin, Poland) and were of analytical grade. Isoflurane was purchased from Primal Healthcare UK Ltd. (Morpeth, UK). All other reagents were of analytical grade and were purchased from Sigma-Aldrich (St. Louis, MO, USA).

### 4.10. Statistics

All the results are expressed as mean ± SD. Comparisons between multiple groups were made using one-way ANOVA followed by Bonferroni’s post-hoc test. In the ANOVA test, *p* < 0.05 was considered statistically significant. Bonferroni’s post-hoc test was considered statistically significant at *p* < 0.05 and was interpreted as follows: * *p* < 0.05, ** *p* < 0.01, *** *p* < 0.001 when comparisons were made with the C group. The correlation coefficients were interpreted according to the guidelines by Colton (1974). Statistical analyses were done using Graph Pad Prism version 5.0 for Windows, Graph Pad software, San Diego, CA, USA. The multivariate analysis of the spectral data was carried out using principal component analysis (PCA) from the Statistica 8 software (StatSoft Inc., Tulsa, OK, USA).

## 5. Conclusions

Our findings identify Colec12 and MARCO as scavenger receptors involved in humoral immune response via the macrophage stimulation of immunoglobulin release. Additionally, our results indicate that SCARB1 is a principal candidate receptor for GSSA-induced immunoglobulin secretion correlated with the decrease of the oxidative stress in CD19+ B plasma cells.

## Figures and Tables

**Figure 1 ijms-20-03627-f001:**
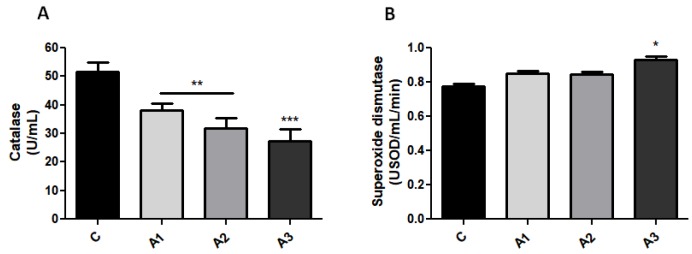
Serum catalase (CAT) and superoxide dismutase (SOD) activity in the control and experimental animals. (**A**) Allicin treatment decreased the catalase activity and (**B**) A3 slowly increased SOD activity. Values are expressed as mean ± SD (* *p* < 0.05; ** *p* < 0.01; *** *p* < 0.001).

**Figure 2 ijms-20-03627-f002:**
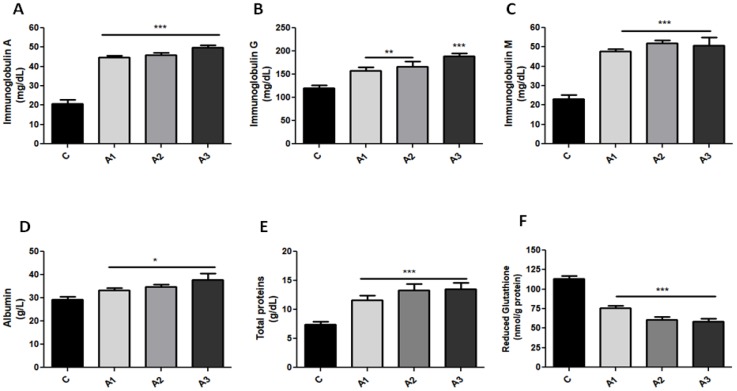
Comparisons of IgA (**A**), IgG (**B**), IgM (**C**), albumin (**D**), total proteins (**E**), and reduced glutathione (**F**) in serum of the control and allicin-treated animals. Allicin treatment induced a prominent increase in immunoglobulins as compared to the control group. Allicin in a dose of 5 mg/kg induced IgG values twice as high as in the control group, whereas albumin and total proteins were also increased. Glutathione pool was decreased after allicin administration. Values are expressed as mean ± SD (* *p* < 0.05; ** *p* < 0.01; *** *p* < 0.001).

**Figure 3 ijms-20-03627-f003:**
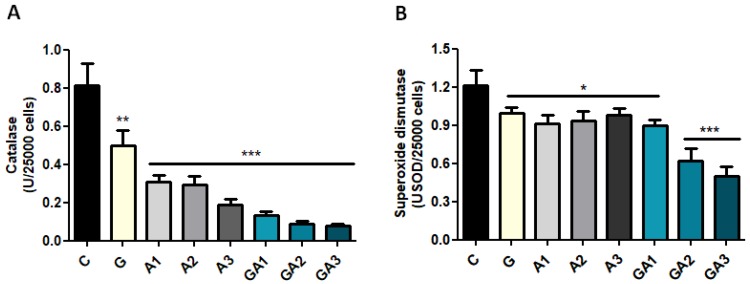
Assessment of CAT (**A**) and SOD (**B**) activity in the CD19+ lymphocytes. Glutathione, allicin and allicin + glutathione (GSSA) significantly decreased CAT and SOD activity in the CD19+ lymphocytes. Values are expressed as mean ± SD (* *p* < 0.05; ** *p* < 0.01; *** *p* < 0.001).

**Figure 4 ijms-20-03627-f004:**
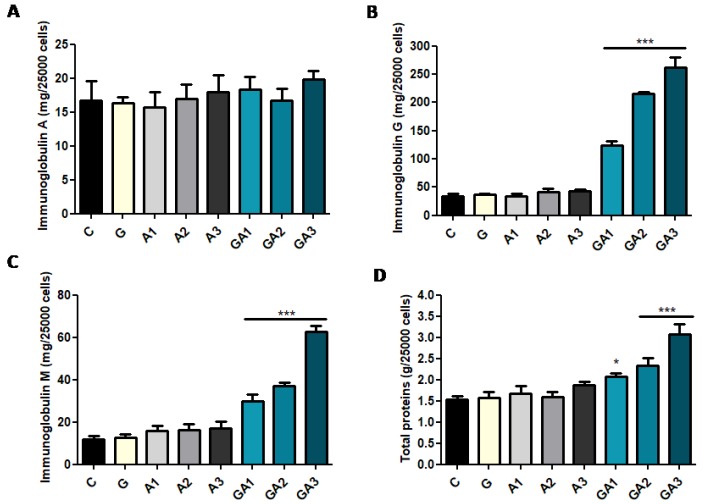
Comparisons of IgA (**A**), IgG (**B**), IgM (**C**) and total proteins (**D**) for the control and allicin-treated groups of CD19+ cells. Allicin and allicin + glutathione treatment did not induce variations in IgA levels. IgG and IgM were increased in allicin + glutathione (GSSA)-treated cells, and the single administration of allicin did not influence IgG and IgM secretion. Values are expressed as mean ± SD (* *p* < 0.05; ** *p* < 0.01; *** *p* < 0.001).

**Figure 5 ijms-20-03627-f005:**
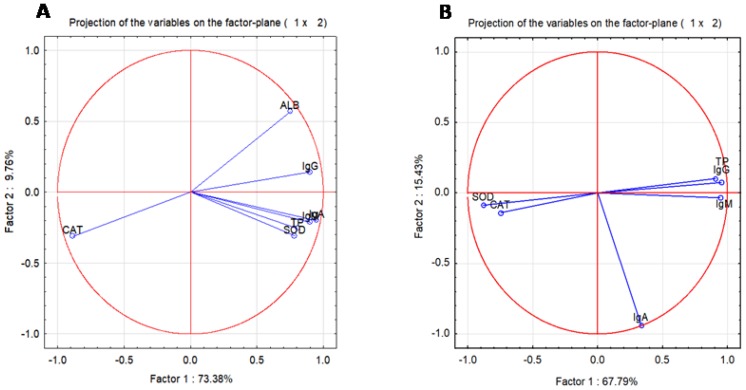
Correlation diagrams between serum (**A**) and lymphocytes parameters (**B**) investigated using the principal component analysis (PCA) model. In serum, a negative correlation between CAT and immunoglobulins and a positive correlation between SOD and immunoglobulins demonstrate that the oxidative stress directly influences antibodies’ secretion, whereas in in vitro tests, CAT and SOD were negatively correlated with antibodies’ secretion. As in the in vivo models, when the oxidative stress is decreased, the immunoglobulin secretion is amplified.

**Figure 6 ijms-20-03627-f006:**
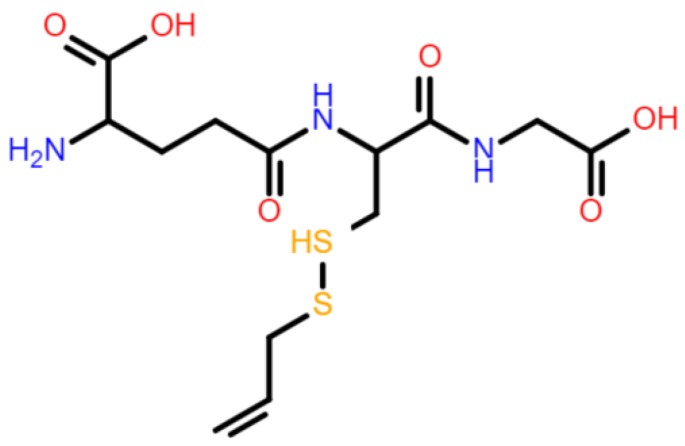
Chemical structure of S-allylmercaptoglutathione drawn in Gauss View Version 5 [31].

**Figure 7 ijms-20-03627-f007:**
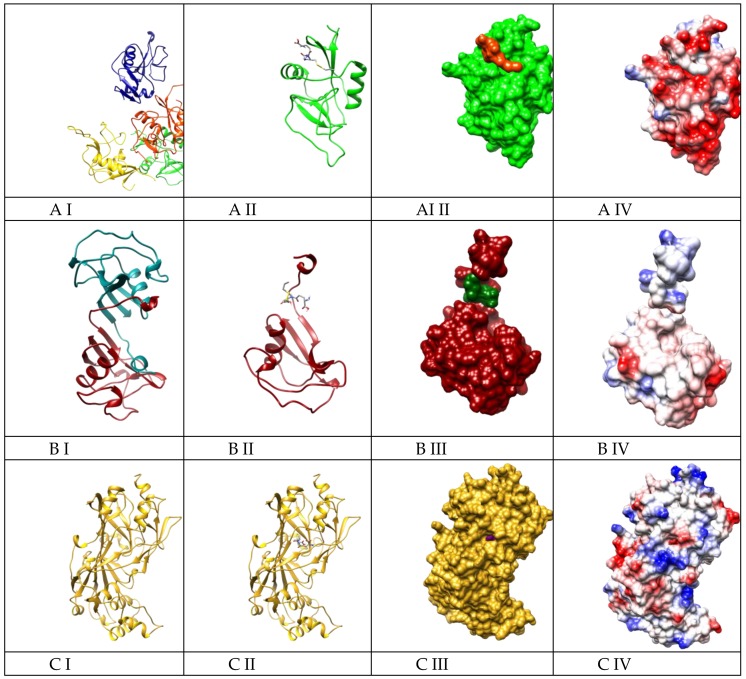
Scavenger receptors’ geometry and models of the GSSA interaction with selected scavenger receptors. *I*: **A I**, Colec12 (quadrimer); **B I**, MARCO (dimer); **C I**, SCARB1. **II**: Geometries of GSSA–SCAR systems after docking; for a better view of the binding site, the figures represent just one monomer for each receptor: **A II**, Colec12 monomer A; **B II**, MARCO monomer A; **C II**, SCARB1. **III**: Surfaces of S-allylmercaptoglutathione – SCAR systems after docking. **IV**: Coulombic surfaces of S-allylmercaptoglutathione – SCAR systems, with the red parts being negatively charged and the blue ones being positively charged.

**Figure 8 ijms-20-03627-f008:**
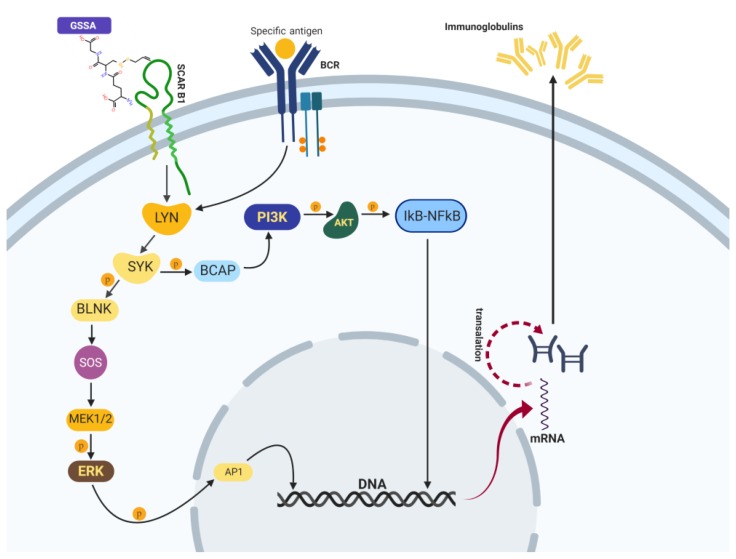
Signaling pathway for GSSA via SCARB1 receptor from CD19+ lymphocyte. According to related studies [54,55] as well as to KEGG databases, the signaling pathway of SCARB1 is based on tyrosine kinase LYC and SYC which activate PI3K-AKT signaling pathway as well as BLNK-MEK1/2-ERK-AP1 pathway and generate DNA transcription with posttranslational folding of IgA, IgG, and IgM. A cross-talk point between SCARB1 and BCR is the activation of PI3K as well as ERK [52]. Figure 8 was created using Premium BioRender software (license related code # 8D9372A0-000, Premium version, Toronto, ON, Canada) and KEGG databases.

**Table 1 ijms-20-03627-t001:** Unchanged hematological parameters in the control animals and in the animals treated with allicin. The administration of allicin did not induce significant changes (*p* > 0.05) in white blood cell parameters.

Parameters	C	A1 Allicin 1.25 mg/kg	A2 Allicin 2.5 mg/kg	A3 Allicin 5 mg/kg
WBC (10^9^/L)	14.5 ± 1.63	13.1 ± 1.23	13.4 ± 1.02	14.2 ± 2.14
LYM (10^9^/L)	10.1 ± 1.49	9.7 ± 1.24	10.5 ± 1.02	11.0 ± 1.76
MON (10^9^/L)	0.3 ± 0.10	0.2 ± 0.08	0.3 ± 0.31	0.2 ± 0.09
NEU (10^9^/L)	2.8 ± 0.13	3.2 ± 0.12	2.7 ± 0.20	2.9 ± 0.45
LYM%	76.4 ± 1.93	73.5 ± 2.80	75.1 ± 2.78	77.8 ± 2.20
MON%	2.08 ± 0.85	1.57 ± 0.97	2.2 ± 2.04	1.30 ± 0.47
NEU%	20.8 ±1.26	25.1 ± 1.73	21.7 ± 3.35	20.9 ± 2.06

**Table 2 ijms-20-03627-t002:** Binding energies of GSSA to scavenger receptors type A (Colec12, MARCO, Msr1, SCARA3, SCARA5) and type B (SCARB1).

Ligand	Type	Binding Energy (kcal/mol)	Geometric Specificity
Colec12	SCAR A	−6.7	Very high
MARCO	SCAR A	−5.7	Very high
Msr1	SCAR A	−4.7	Low
SCARA3	SCAR A	−4.8	Low
SCARA5	SCAR A	−5.0	Very low
SCARB1	SCAR B	−5.7	Very high

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
