# Peer review of "New Aspects Towards a Molecular Understanding of the Allicin Immunostimulatory Mechanism via Colec12, MARCO, and SCARB1 Receptors"

_ijms, 2019, doi:10.3390/ijms20153627_

Round 1

Reviewer 1 Report

Review

 In this paper, the authors investigated the immunostimulatory effect of allicin in vivo in Wistar rats as well as in vitro on CD19+ lymphocytes. They identified two scavenger receptors class A (Colec12 and MARCO) and one receptor class B (SCARB1) involved in the humoral immune response via the macrophage stimulation. The immunostimulatory properties of garlic and allicin are well known but the mechanism of their action has not been fully elucidated, yet. Thus, the problem addressed by the authors is interesting and up-to-date, however, the paper is not well written and it requires a thorough improvement.

 Major points:Some sentences do not make sense or are grammatically incorrect, e.g. the last paragraph in the introduction. A check of the text by a native speaker is strongly recommended.Results-In the methods section, the authors wrote that antioxidant enzymes and GSH were assayed in serum, however, in Figure 1 legend, blood CAT and SOD activity is presented. What was the sample – whole blood or serum?-It should be clearly stated also in the Figure 2 legend what kind of body fluid was used (blood or serum). What does it mean: “Reduced glutathione in Control and allicin-treated animals”?-In my opinion the level of GSH presented in Figure 2 is incorrect regardless of whether it is GSH level in serum or blood. Normal GSH level in the cells is 1-10 mM and its plasma or serum level is much lower. The level of GSH in control animals presented in Figure 2 is 1000 μmol/ml, what means 1 mol per liter (1 M)!-The authors presented the results of albumin concentration assay, which in control rats was 29.2 g/L, while normal range of albumin (as they reported) is 37-58 g/L. Why was the level of albumin in control rats lower than the normal level?-The authors wrote that their “data exhibit a prominent antioxidant effect of allicin, marked by decreasing of catalase activity” – why does a decrease in the activity of one of the main antioxidant enzymes mean an antioxidant effect? Why is a decrease in CAT considered by the authors as downregulation of oxidative stress? (page 3, line 104). -Page 3, line 103 – SOD activity in the control group is 0.81 USOD/ml/min in the text, while in the Figure 1B it is lower than 0.8 USOD/ml/min.-Page 3, line 105 - What is IgA/CAT? Why did the authors use this parameter? It should be clarified.The paragraph (page 3, lines 104-108) should be included after all results presented in Figures 1-2 (at the end of paragraph 2.1) have been described.-The experiment performed by the Authors revealed a decrease in CAT activity upon the allicin treatment, while SOD activity was increased in A3 group of animals. It seems to be contradictory because as a result of SOD activity hydrogen peroxide is formed, which is a substrate for catalase. The authors should discuss this fact.-Paragraph 2.3, line 153 – it should be specified were an increase in immunoglobulins and a decrease in catalase activity was observed (in blood/serum of animals?)-Page 4, line 159 – “Allicin and … decreased CAT activity” – where? – in CD19+?-It is not acceptable to express the activity CAT or SOD in CD19+ cells in U/ml (the same applies to IgA in mg/dL – what does it mean mL or dL of cells? It should be presented in U (or mg) per the number of cells.-Figure 4 – there is no information in the legend where these results were obtained. I guess that in CD19+ cells.-Figure 5 – the correlation diagram – there is no information which diagram is A, which B. Normally, the left panel is designated A – if it is true, the information is wrong because only CAT, not the antioxidant enzymes are negatively correlated.-Page 7, lines 208-209 – in this sentence, in vitro (or in CD19+ cells) should be added.-Page 7, lines 211-212 – this sentence is the same as the previous one; it is not “in addition”.-Paragraph 2.4 – The authors wrote that three scavenger receptors: Colec12, MARCO as well as SCARB1 have high interaction energy and good geometrical match, so why the title of the paper is “Towards a Molecular Understanding of the Allicin Immunostimulatory Mechanism   via B Scavenger Receptors Type 1”? I suggest to modify the title (maybe “new aspects of ..”-In the concluding remarks, the authors speak of the mechanism of oxidative stress reduction (in my opinion oxidative stress was not assayed in this study) via PI3K and ERK - it is only a speculation, the authors did not study it.  The link of SCARB1 with PI3K and ERK is mentioned only in Figure 8 legend. I suggest to explain it better in the discussion.  MethodsThis section should be rearranged. Allicin and other chemicals can be mentioned in one paragraph. The mention about allicin degradation products in this paragraph is unnecessary. The authors wrote that the animals were divided into 4 equal groups but there is no information about the number of rats in the control and experimental group. How much blood was taken from one rat? Why were female rats used in this experiment? How long after the last dose of treatment were the animals euthanized? Animals, treatment and ethics statement should be given before the estimated parameters (Immunoglobulins, oxidative stress etc). Immunoglobulins, total proteins, CAT, GSH were assayed in the rat serum as well as in CD19+  cells – it should be clearly stated in the methods section. In the section 4.6 (in vitro) there is no information about the origin of blood used for PBMC isolation (animal or human).   Minor points:Are the initials of Adrian Bogdan Tigu (A.D.F) and Anka D. Farcas (A.B.T) correct? It looks like they are swapped.In the affiliation 6 and 7 the e-mail address is: [email protected]Line 382 -  “Animals were maintained … under 230C” – it should be replaced by 230CLine 408 – “40C” – it should be replaced by 40CPage 3, line 100 – p<001 should be replaced by p<0,01The unit μMol/mL is not acceptable, it should be replaced by μmol/mlFigure 3 legend – “CAT and SOD activity in the..?”Page 12, line 415 - ‘The animals were euthanized by deer deep prolonged narcosis and were considered death when no respiratory and heart activity was recorded” – what does “deer” mean here?

Author Response

Major points:

1. Some sentences do not make sense or are grammatically incorrect, e.g. the last paragraph in the introduction. A check of the text by a native speaker is strongly recommended.

R: Thank you! The manuscript was revised by our colleague Mihail Buse, researcher at Medfuture Centere for Advanced Medicine, a native Canadian and native English speaker.

Results

2. In the methods section, the authors wrote that antioxidant enzymes and GSH were assayed in serum, however, in Figure 1 legend, blood CAT and SOD activity is presented. What was the sample – whole blood or serum?

R: The serum was used for the mentioned parameters in Figure 1 (line 122).

3. It should be clearly stated also in the Figure 2 legend what kind of body fluid was used (blood or serum).

R: Also, the serum was used for the determination of the parameters mentioned in Figure 2 (line 127).

4. What does it mean: “Reduced glutathione in Control and allicin-treated animals”?-In my opinion the level of GSH presented in Figure 2 is incorrect regardless of whether it is GSH level in serum or blood. Normal GSH level in the cells is 1-10 mM and its plasma or serum level is much lower. The level of GSH in control animals presented in Figure 2 is 1000 μmol/ml, what means 1 mol per liter (1 M)!

R: Thank you for this observation! During the revision time, using samples which were kept at -600 C we repeated the GSH assay by the method proposed by Tietze (1969) and GSH concentration was expressed in nmol/g protein; as in the Elmman method, DTNB is the principal reagent but in Tietze method deproteinization was done with trichloroacetic acid not with absolute methanol as in the first method that was used in our research; the measure unit of GSH is nmol/g protein. Thank you very much for your pertinent and well formulated observation! [C, 113.3 ± 6.87 nmol/g protein, A1, 75.5 ± 7.17 nmol/g protein, A2, 60.7 ± 7.52 nmol/g protein, A3, 58.3 ± 7.51 nmol/g protein].

5. The authors presented the results of albumin concentration assay, which in control rats was 29.2 g/L, while normal range of albumin (as they reported) is 37-58 g/L. Why was the level of albumin in control rats lower than the normal level?

R: The albumin concentration in Control rats was lower than the reference range because in rats, albumin varies with age, hormonal fluctuations as well as with the degree of hydration [27, 28]. Allicin increased the albumin concentration with values that remained in normal range (lines 113-116).

6. The authors wrote that their “data exhibit a prominent antioxidant effect of allicin, marked by decreasing of catalase activity” – why does a decrease in the activity of one of the main antioxidant enzymes mean an antioxidant effect? Why is a decrease in CAT considered by the authors as downregulation of oxidative stress? (page 3, line 104). 

R: In this context, decreasing CAT activity can be related with a lowered oxidative stress because CAT was decreased in both serum and cells cultures and SOD was slight increased in serum and decreased in cells cultures. Catalase is a common factor for our in vivo and in vitro tests and CAT decreasing was related, as a single indicator, to antioxidant effect of the allicin. Furthermore, as a second-line antioxidant enzyme, CAT decreasing follows after GPX intervention in peroxide scavenging [38] thus CAT decrease demonstrates the overall decrease of the oxidative stress.

7. Page 3, line 103 – SOD activity in the control group is 0.81 USOD/ml/min in the text, while in the Figure 1B it is lower than 0.8 USOD/ml/min.

R: Indeed, SOD activity in Control was 0.77 ± 0.02 USOD/mL/min; all values of all parameters were verified and were in accordance with data and graphics. Thank you for the observation!

8. Page 3, line 105 - What is IgA/CAT? Why did the authors use this parameter? It should be clarified.

R: IgA/CAT is in fact IgA versus CAT in correlation analysis; this aspect was modified in IgA vs. CAT, IgM vs. CAT, IgG vs. CAT all of these comparisons being associated with R and p.

9. The paragraph (page 3, lines 104-108) should be included after all results presented in Figures 1-2 (at the end of paragraph 2.1) have been described.

R: This change was done (lines 132-136).

10.The experiment performed by the Authors revealed a decrease in CAT activity upon the allicin treatment, while SOD activity was increased in A3 group of animals. It seems to be contradictory because as a result of SOD activity hydrogen peroxide is formed, which is a substrate for catalase. The authors should discuss this fact.

R: CAT and SOD decreased in their activities almost simultaneously however we have noticed these two enzymes (SOD and CAT) do not follow the same behavior as expected. SOD is an inducible and polymorphic enzyme that can be activated together with GPX by antioxidants via Nrf2 signaling [39] therefore SOD increasing in serum in A3 group suggests that allicin in high concentration can activate this enzyme. The apparent contradiction between serum SOD increasing and CAT decreasing reveals the polifactorial feature of the antioxidant response in in vivo systems whereas in CD19+ cells SOD and CAT have varied in tandem. SOD and CAT was decreased in CD19+ cells after 24h of allicin and allicin-GSH exposure because superoxide production was balanced by direct allicin scavenging effect [1, 6, 11] and peroxide concentration was also reduced. In this context, the major difference between serum and CD19+ cells is that in in vivo system occur complementary antioxidant factors (nonenzymatic antioxidants, hormonal regulation) while CD19+ cells cultures are an isolated system.

11. Paragraph 2.3, line 153 – it should be specified were an increase in immunoglobulins and a decrease in catalase activity was observed (in blood/serum of animals?)

R: Line 157, in serum.

12. Page 4, line 159 – “Allicin and … decreased CAT activity” – where? – in CD19+?

R: Line 163, in CD19+ cells.

13. It is not acceptable to express the activity CAT or SOD in CD19+ cells in U/ml (the same applies to IgA in mg/dL – what does it mean mL or dL of cells? It should be presented in U (or mg) per the number of cells.

R: These changes were done in all Figures that require this change.

14. Figure 4 – there is no information in the legend where these results were obtained. I guess that in CD19+ cells.

R: Line 173, yes, in CD19+ cells.

15. Figure 5 – the correlation diagram – there is no information which diagram is A, which B. Normally, the left panel is designated A – if it is true, the information is wrong because only CAT, not the antioxidant enzymes are negatively correlated.

R: Thank you for the observation; Figure 5 is now with A (left) and B (right) and indeed, just CAT is negatively correlated; this change was done in line 203 and in Figure 5.

16. Page 7, lines 208-209 – in this sentence, in vitro (or in CD19+ cells) should be added.

R: This change was done (now is line 212).

17. Page 7, lines 211-212 – this sentence is the same as the previous one; it is not “in addition”.

R: This change was done in the lines 213-217 (the paragraph was revised).

18. Paragraph 2.4 – The authors wrote that three scavenger receptors: Colec12, MARCO as well as SCARB1 have high interaction energy and good geometrical match, so why the title of the paper is “Towards a Molecular Understanding of the Allicin Immunostimulatory Mechanism   via B Scavenger Receptors Type 1”? I suggest to modify the title (maybe “new aspects of ..”)

R: The title was modified in: ``New aspects towards molecular understanding of the allicin immunostimulatory mechanism via Colec12, MARCO and SCAR B1 receptors``

19. In the concluding remarks, the authors speak of the mechanism of oxidative stress reduction (in my opinion oxidative stress was not assayed in this study) via PI3K and ERK - it is only a speculation, the authors did not study it.  The link of SCARB1 with PI3K and ERK is mentioned only in Figure 8 legend. I suggest to explain it better in the discussion. 

R: ERK and PI3K were cut from the Concluding remarks and in Discussion, according to your suggestion, the relation between SCAR B1 and PI3K-ERK was explained better (line 364 - line 372): ``Signaling through SCAR B1 interferes with the signaling pathway of BCR, a specific receptor involved in immunoglobulin secretion via tyrosine kinase LYN and SYK. However, SYK tyrosine kinase can direct the signaling cascade either to enable PI3K or activate ERK. The coactivation of PI3K-ERK is also a possible cross-talk signaling pathway of SCAR B1 - BCR [55] in B lymphocytes. In immunoglobulin secretion via BCR, LYN/SYK - ERK is the main signaling pathway whereas costimulation by CD19 was associated with PI3K activation [55, 56]. SCAR B1 signaling becomes overlapped with BCR signaling cascade in tyrosine kinase LYN-SYK which determines the next reactions (via PI3K and ERK) for immunoglobulin secretion. ``

Methods

20. This section should be rearranged. Allicin and other chemicals can be mentioned in one paragraph.

R: These changes were done (line 470-475).

21. The mention about allicin degradation products in this paragraph is unnecessary. 

R: These data were deleted.

22. The authors wrote that the animals were divided into 4 equal groups but there is no information about the number of rats in the control and experimental group.

R: The animals were divided into 4 equal groups (5 animals/group) (line 392- line 393).

23. How much blood was taken from one rat?

R: For hematological evaluation 200 µL of blood were collected from retro-orbital plexus (line 409) and then, for biochemical determinations, 2 mL of blood were collected in clot activated vacutainers (line 413).

24. Why were female rats used in this experiment?

R: We used female rats because we evaluated in another experiment the effect of allicin on the development and our data must be obtained from the same gender because these researches are complementary.

25. How long after the last dose of treatment were the animals euthanized?

R: After 12h (line 407).

26. Animals, treatment and ethics statement should be given before the estimated parameters (Immunoglobulins, oxidative stress etc).

R: These changes were done.

27. Immunoglobulins, total proteins, CAT, GSH were assayed in the rat serum as well as in CD19+ cells – it should be clearly stated in the methods section. 

R: These changes were done: line 416, line 418-419, line 420.

28. In the section 4.6 (in vitro) there is no information about the origin of blood used for PBMC isolation (animal or human).  

R: Human Peripheral blood mononuclear cells (PBMC) were isolated from fresh peripheral whole blood (line 423).

Minor points:

29. Are the initials of Adrian Bogdan Tigu (A.D.F) and Anka D. Farcas (A.B.T) correct? It looks like they are swapped.

R: Indeed. I changed these aspects in manuscript (line 8-9).

30. In the affiliation 6 and 7 the e-mail address is: [email protected]

R: This email is [email protected] (line 20).

31. Line 382 -  “Animals were maintained … under 230C” – it should be replaced by 230CLine 408 – “40C” – it should be replaced by 40C.

R: These changes were done (line 399, line 448).

32. Page 3, line 100 – p<001 should be replaced by p<0,01. The unit μMol/mL is not acceptable, it should be replaced by μmol/ml.

R: These changes were done in all paragraphs and Figures which have required these modifications (line 99) and μMol/mL was changed in according to calculation method in nnmol/g protein (for GSH concentration).

33. Figure 3 legend – “CAT and SOD activity in the..?”

R: Assessment of (A) CAT and (B) SOD activity in the CD19+ lymphocytes (line 174).

34. Page 12, line 415 - ‘The animals were euthanized by deer deep prolonged narcosis and were considered death when no respiratory and heart activity was recorded” – what does “deer” mean here?

R: Deer was an error (repeated words); deep narcosis is a current form (line 405). Thank you!

Reviewer 2 Report

This study analyzes the antioxidant and immunostimulatory effects of allicin. This compound may explain at least part of the bioactive properties of some plant extracts. The results suggests that the in vivo effects may be produced by a derivate with glutathione, and the immunostimulation by direct activation of B lymphocytes mediated by binding to sacavenger receptors.

Methods

¿How was the compound administered? ¿By gavage or with the food?

Results

The numerical data in the text makes it difficult to read. In most cases it could be deleted, as the same information is expressed in the figures.

There is an apparent contradiction in that SOD is increased in vivo but reduced in vitro. This should be discussed.

Concluding remarks

Although it is likely that the effects may be mediated by the ERK and PI3K pathways, this has not been studied in this work, so it should not be included in the conclusion.

Author Response

Methods

1. How was the compound administered? By gavage or with the food?

R: Allicin was administered via gavage (line 394-395).

Results

2. The numerical data in the text makes it difficult to read. In most cases it could be deleted, as the same information is expressed in the figures.

R: Indeed, thank you for your observation. However, Rev#1 made some observations related to numerical data and some of the comments are based on these data. Therefore, I stay between two situations: if I delete the numerical data from the text, some of the responses to Rev#1 are not available and on the other side, we want to improve the quality of the manuscript taking into consideration all suggestions from the reviewers.

3. There is an apparent contradiction in that SOD is increased in vivo but reduced in vitro. This should be discussed.

R: Yes. This apparent contradiction was discussed in lines 288-305:

In this context, decreasing CAT activity can be related with a lowered oxidative stress because CAT was decreased in both serum and cells cultures and SOD was slight increased in serum and decreased in cells cultures. Catalase is a common factor for our in vivo and in vitro tests and CAT decreasing was related, as a single indicator, to antioxidant effect of the allicin. Furthermore, as a second-line antioxidant enzyme, CAT decreasing follows after GPX intervention in peroxide scavenging [38] thus CAT decrease demonstrates the overall decrease of the oxidative stress. CAT and SOD decreased in their activities almost simultaneously however we have noticed these two enzymes (SOD and CAT) do not follow the same behavior as expected. SOD is an inducible and polymorphic enzyme that can be activated together with GPX by antioxidants via Nrf2 signaling [39] therefore SOD increasing in serum in A3 group suggests that allicin in high concentration can activate this enzyme. The apparent contradiction between serum SOD increasing and CAT decreasing reveals the polifactorial feature of the antioxidant response in in vivo systems whereas in CD19+ cells SOD and CAT have varied in tandem. SOD and CAT was decreased in CD19+ cells after 24h of allicin and allicin-GSH exposure because superoxide production was balanced by direct allicin scavenging effect [1, 6, 11] and peroxide concentration was also reduced. In this context, the major difference between serum and CD19+ cells is that in in vivo system occur complementary antioxidant factors (nonenzymatic antioxidants, hormonal regulation) while CD19+ cells cultures are an isolated system.

Concluding remarks

4. Although it is likely that the effects may be mediated by the ERK and PI3K pathways, this has not been studied in this work, so it should not be included in the conclusion.

R: Indeed. Thank you for the observation! ERK and PI3K were deleted and Concluding remarks are as follow (lines 488-491):

Our findings indentify Colec12 and MARCO as scavenger receptors involved in humoral immune response via the macrophage stimulation of immunoglobulin releasing. Also, our results indicate that SCARB1 is a principal candidate receptor for SAMG-induced immunoglobulin secretion correlated with the decrease of the oxidative stress in CD19+ B plasma cells.

Thank you for all observations!

Round 2

Reviewer 1 Report

The revised version of the manuscript is now much better, however, it still needs some slight corrections:

There are still grammar mistakes (i.e. “reagent were used”; “serum catalase are..”)

In all Figure legends and in the text, there should be: Comparison, assessment or correlation of sth (IgA, CAT, SOD, etc.) and then (A), (B), etc. It is not good: Assessment of (A) CAT and (B) SOD activity.

Figure 5 has now 4 diagrams, the first two should be deleted. Figure 5 legend – Correlations diagram between in serum (A) and cell cultures (B) parameters  - this sentence is not proper and should be corrected.

Page 12, lines 384-385 – this sentence is unclear. I think that CD19+ cells were stimulated by allicin and SAMG, not “costimulation by CD19 was associated”

Page 14, line 415 – there is no space between words

Page 15, line 443 – “peripheral” instead “Peripheral”

Figure legends:

All descriptions should be corrected as was mentioned earlier

Figure 3 – the sentence in not finished … in the.”

There are no spaces after some figure numbers and descriptions.

Figure 6 – the molecular mass expressed in daltons is used mainly for proteins not to low molecular weight compounds, so this information is not useful. Moreover, 319 g/mol is not 0,19 kDa!

Figure 8 – the abbreviations are LYC and SYC, not Lyc and Syc.

Author Response

1. There are still grammar mistakes (i.e. “reagent were used”; “serum catalase are...”)

A: line 237 and line 322 contain corrected forms (was used and serum catalase is…); the blue words are for cursive and correct language.

2. In all Figure legends and in the text, there should be: Comparison, assessment or correlation of sth (IgA, CAT, SOD, etc.) and then (A), (B), etc. It is not good: Assessment of (A) CAT and (B) SOD activity.

A: The changes were done: lines 200-201, 221, 287, 314-315, 325

3. Figure 5 has now 4 diagrams, the first two should be deleted. Figure 5 legend – Correlations diagram between in serum (A) and cell cultures (B) parameters - this sentence is not proper and should be corrected.

A: First 2 diagrams were deleted. The correction was done in the lines 325-326: Correlation diagrams between serum (A) and lymphocytes (B) parameters investigated using the PCA model.

4. Page 12, lines 384-385 – this sentence is unclear. I think that CD19+ cells were stimulated by allicin and SAMG, not “costimulation by CD19 was associated”

A: Lines 608-609: In immunoglobulin secretion via BCR, LYN/SYK - ERK is the main signaling pathway, in comparison to the CD19+ cells where stimulation by allicin and SAMG was associated with PI3K activation [55, 56].

5. Page 14, line 415 – there is no space between words

A: The correction was applied.

6. Page 15, line 443 – “peripheral” instead “Peripheral”

A: Done (line 680)

Figure legends: 

7. All descriptions should be corrected as was mentioned earlier

A: Done lines 200-201, 221, 287, 314-315, 325

8. Figure 3 – the sentence in not finished … in the.”

A: line 288-289 … in the CD19+ lymphocytes.

9. There are no spaces after some figure numbers and descriptions.

A: The corrections were applied.

10. Figure 6 – the molecular mass expressed in daltons is used mainly for proteins not to low molecular weight compounds, so this information is not useful. Moreover, 319 g/mol is not 0,19 kDa!

A: This information was deleted (line 366).

11. Figure 8 – the abbreviations are LYC and SYC, not Lyc and Syc.

A: The correction was applied (line 624)